**Data Availability Statement:** The data underlying the results presented in the study have been provided as supporting information S1 Data.

**Funding:** The author(s) received no specific funding for this work.

# Correlates of tuberculosis in children under the age of 15 years

**Lukundo Siame** , **Gift C. Chama** , **Sepiso K. Masenga** *

School of Medicine and Health Sciences, Mulungushi University, Livingstone, Zambia

* sepisomasenga@gmail.com, smasenga@mu.ac.zm

## Abstract

### Background

Tuberculosis (TB) remains a significant public health challenge, particularly among vulnerable populations like children. This is especially true in Sub-Saharan Africa, where the burden of TB in children is substantial. Zambia ranks 21st among the top 30 high TB endemic countries globally. While studies have explored TB in adults in Zambia, the prevalence and associated factors in children are not well documented. This study aimed to determine the prevalence and sociodemographic, and clinical factors associated with active TB disease in hospitalized children under the age of 15 years at Livingstone University Teaching Hospital (LUTH), the largest referral center in Zambia's Southern Province.

### Methods

This retrospective cross-sectional study of 700 pediatric patients under 15 years old, utilized programmatic data from the Pediatrics Department at LUTH. A systematic sampling method was used to select participants from medical records. Data on demographics, medical conditions, anthropometric measurements, and blood tests were collected. Data analysis included descriptive statistics, chi-square tests, and multivariable logistic regression to identify factors associated with TB.

### Results

The median age was 24 months (interquartile range (IQR): 11, 60) and majority were male (56.7%, n = 397/700). Most participants were from urban areas (59.9%, n = 419/700), and 9.2% (n = 62/675) were living with HIV. Malnutrition and comorbidities were present in a significant portion of the participants (19.0% and 25.1%, respectively). The prevalence of active TB cases was 9.4% (n = 66/700) among hospitalized children. Persons living with HIV (Adjusted odds ratio (AOR) of 6.30; 95% confidence interval (CI) of 2.85, 13.89, p< 0.001), and those who were malnourished (AOR: 10.38, 95% CI: 4.78, 22.55, p< 0.001) had a significantly higher likelihood of developing active TB disease.

### Conclusion

This study revealed a prevalence 9.4% active TB among hospitalized children under 15 years at LUTH. HIV status and malnutrition emerged as significant factors associated with

**Competing interests:** The authors have declared that no competing interests exist.

active TB disease. These findings emphasize the need for pediatric TB control strategies that prioritize addressing associated factors to effectively reduce the burden of tuberculosis in Zambian children.

## Introduction

Tuberculosis (TB) remains a significant public health challenge globally, particularly among vulnerable populations such as children [1]. According to the World Health Organization (WHO), an estimated 10.6 million people worldwide developed active TB in 2022, with children under the age of 15 years accounting for 1.3 million of all TB cases and TB being among the top causes of mortality in children [2].

The burden of TB in Africa is high, in 2022 one-third (320,000) of all TB notifications worldwide were among children between 0 to 15 years from Africa, with a large number who remained undiagnosed or unreported [3]. The high burden has been attributed to a number of factors such as malnutrition, diabetes, and HIV infection [4, 5]. Barriers to treatment initiation, including limited knowledge, attitudes, and beliefs about tuberculosis, as well as economic burdens, have been identified [3, 6]. The high prevalence of TB in children in this region underscores the need for improved access to healthcare and the removal of treatment barriers [7].

The burden of TB in Zambia is high as it is ranked 21$^{st}$ among the top 30 high-burden countries [8]. The incidence rate of TB in Zambia was estimated to be about 295 per 100,000 population per year, and 9% of TB cases occur in children, slightly below the 10% benchmark by WHO in 2022 [9, 10]. However, the magnitude of TB in the pediatric population is not well studied in Zambia [11]. Thus, understanding the prevalence of TB among children in this setting is crucial for guiding effective public health interventions and resource allocation.

The aim of this study was firstly to determine the prevalence of TB among children under the age of 15 and secondly, to identify sociodemographic and clinical factors associated with TB at Livingstone University Teaching Hospital (LUTH) pediatric department.

## Methods

### Study design and setting

This was a retrospective cross-sectional study where we abstracted secondary data from the Pediatrics Department at LUTH among hospitalized children under the age of 15 years. Data was collected for the period between January 1$^{st}$, 2020, and December 31$^{st}$ 2022 (a three-year period). The department admits about 1000 to 1500 annually to the general ward and malnutrition ward. It also has a High Dependency Unit (HDU), and a Neonatal Intensive Care Unit (NICU). LUTH is the largest referral center in the Southern Province of Zambia.

### Sample size

The sample size calculation was based on a study conducted among inpatient children who died and underwent a postmortem at the University Teaching Hospital in Zambia, which revealed a prevalence of 8.0% for TB [12], with confidence limits of 2.1%, we estimated the minimum sample size required to be 641 with confidence level set at 95%. We used openepi. com to estimate the sample size and abstracted a total of 700 files.

### Eligibility and sampling method

Clinical data was abstracted from the medical records of patients. Only participants who were under the age of 15 years were included. Records with missing data on the outcome variable and age were excluded. To select participants, a systematic sampling method was used, choosing every fourth record. We divided the estimated total number of files in the department (n = 3092) by the sample size to generate a sampling frame (6±2). All 700 selected records (**Fig 1**) were then entered into research electronic data capture (REDCap), a data management tool.

### Variables

The main outcome was tuberculosis, and the independent variables considered were demographics (Age, Sex, Residence), medical conditions (HIV Status, Malnutrition, Comorbidities), anthropometric measurements (Weight(kg) and Height (cm)) and complete blood count (Platelet count, Neutrophil count, Lymphocyte count, Monocyte count, Hemoglobin, White blood cell count). Tuberculosis was diagnosed through laboratory tests or history of receiving treatment for tuberculosis infection aided by chest radiography reports. Additionally, individuals with a measurement of less than -2 standard deviations on growth charts or those

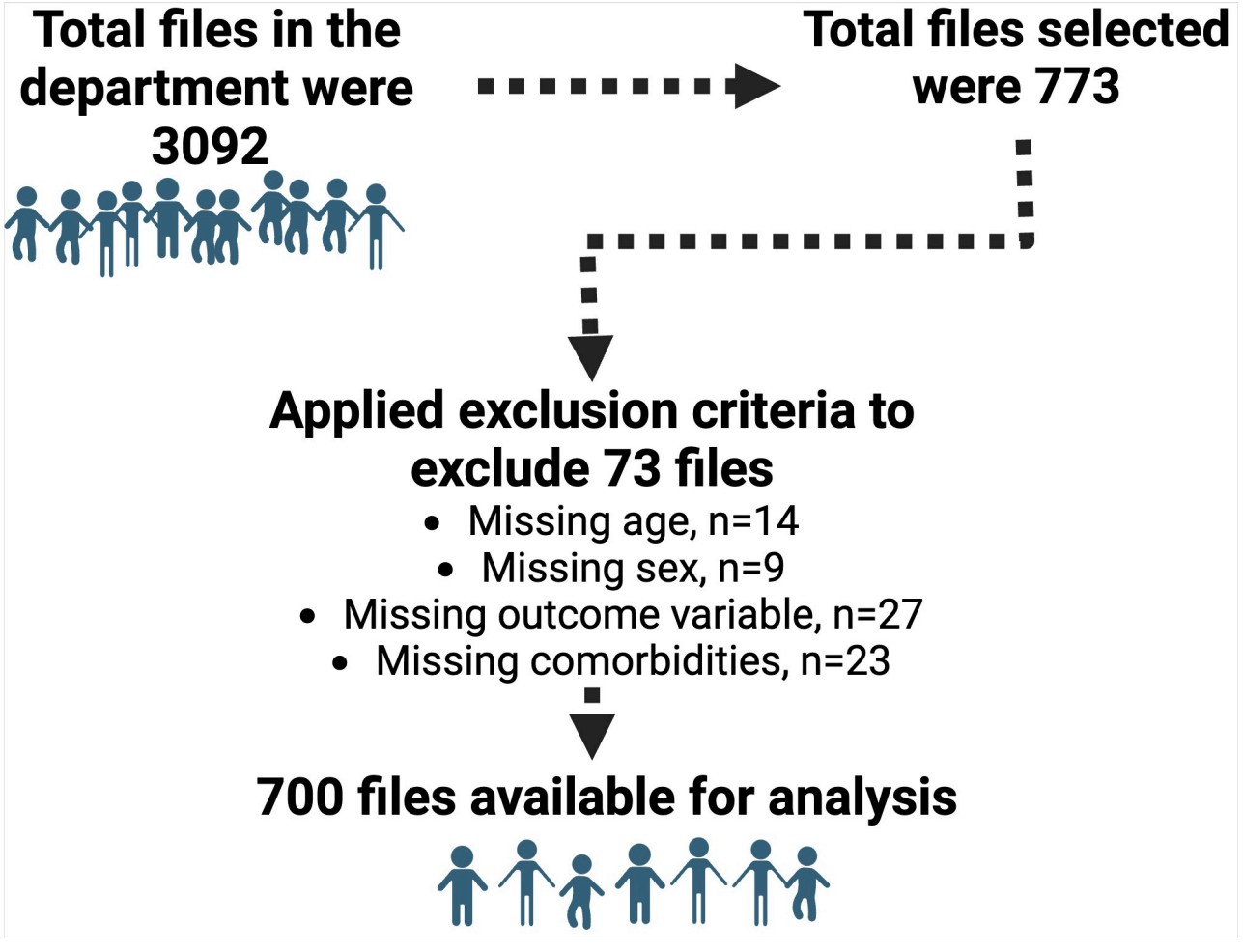

**Fig 1. Data selection and abstraction.**

receiving standard treatment for malnutrition as recommended by the World Health Organization (WHO) were classified as malnourished.

## Data collection

Data were collected by three (3) trained research assistants who abstracted data from patient files between August 1, 2023 and January 8th, 2024. Accuracy of data entered was validated by two (2) senior data abstractors at the end of each day when data abstraction was completed.

## Data analysis

Data were entered in REDCap application first, then exported into excel. Data were then cleaned and thereafter exported to STATA version 15 for Data analysis. To test for normality, histograms were used, followed by Shapiro wilk's test. Data was described using frequency and percentages for categorical variables and median (interquartile range) was used to describe continuous variables. The Wilcoxon rank-sum test was used to compare the statistical difference between the two medians. A relationship between two categorical variables was determined using a chi-squared test. Multivariable logistic regression was used to examine the correlates of tuberculosis using forward stepwise Generalized Linear Models equations (GLM) via the logit link function to account for correlation structure within observations and to select the most significant predictors for inclusion in the final model. To ascertain statistical significance, a p < value of 0.05 was used.

## Ethics

Ethical approval to conduct this study was obtained from Mulungushi University Research Ethics Committee (SMHS-MU2-2023-64) on 6th June 2023. All data analyzed were de-identified to ensure confidentiality. No patient identifying information was abstracted and entered in the data collection form such that no participant was identified during or after the data collection process. Written consent/assent for each participant was not applicable and therefore waived because the data to be collected was secondary.

We have used the Strengthening the Reporting of Observational Studies in Epidemiology (STROBE) for reporting this observational study (see Supporting information file 1).

# Results

## Basic characteristics of participants

The median age of the participants were 24 months (interquartile range (IQR):11, 60) and 56.7% (n = 397) were male. The majority of participants were from urban areas, accounting for 59.9% (n = 419). Persons living with HIV and those who were classified as malnourished were 9.2% (n = 62/675) and 19.0% (n = 133), respectively. Additionally, 25.1% (n = 176) of participants had comorbidities.

## Relationship between active TB disease with sociodemographic and clinical characteristics of participants

The prevalence of TB was 9.4% (n = 66), **Table 1**. TB was more prevalent among patients from rural areas compared to urban areas (53.0% vs. 47.0%). TB was significantly associated with living with HIV compared to those without HIV (43.1% vs. 5.6%). Significant associations were found between malnutrition and TB (71.2% vs. 13.6%). Persons with TB were shorter in height (76.8 cm vs. 83.8 cm), and weighed less than those without TB (7.6 kg vs. 10.0 kg), respectively. TB was more prevalent among children with comorbidities compared to those without

**Table 1. Sociodemographic and clinical characteristics of the study participants.**

| Variable | Median, (IQR) OR Frequency (%) | Tuberculosis | | P-value |
|---|---|---|---|---|
| | | Yes (n = 66, 9.4%) | No (n = 634, 90.6%) | |
| **Age, months** | 24 (11, 60) | 20 (12, 46) | 24 (11, 60) | 0.734 |
| **Sex** | | | | |
| Male | 397 (56.7) | 36 (54.6) | 361 (56.9) | 0.709 |
| Female | 303 (43.3) | 30 (45.4) | 273 (43.1) | |
| **Residence** | | | | |
| Urban | 419 (59.9) | 31 (47.0) | 388 (61.2) | < **0.025** |
| Rural | 281(40.1) | 35 (53.0) | 246 (38.8) | |
| HIV Status, *n = 675* | | | | |
| Living with HIV | 62 (9.2) | 28 (43.1) | 34 (5.6) | <**0.001** |
| HIV negative | 613 (90.8) | 37 (56.9) | 576 (94.4) | |
| **Malnutrition** | | | | |
| Yes | 133 (19.0) | 47 (71.2) | 86 (13.6) | <**0.001** |
| No | 567 (81.0) | 19 (28.8) | 548 (86.4) | |
| **Comorbidities*** | | | | |
| Yes | 176 (25.1) | 34 (51.5) | 142 (22.4) | <**0.001** |
| No | 524 (74.9) | 32 (48.5) | 492 (77.6) | |
| **Anthropometric measurement** | | | | |
| Weight, *kgs* | 9.8 (7.4, 15.5) | 7.6 (5.8,10.6) | 10.0 (7.5,16.0) | <**0.001** |
| Height, *cm* | 82.8 (72.7, 104.2) | 76.8 (69, 86) | 83.8 (73.5,105) | **0.002** |
| Platelet count, *10^9/L* | 320 (220, 425) | 379 (203,551) | 317 (220, 418) | 0.212 |
| Neutrophil count, *10^9/L* | 4.6 (2.8, 8.2) | 3.4 (2.27, 7.1) | 4.8 (2.9,8.4) | 0.196 |
| Lymphocyte count, *10^9/L* | 3.39 (2.2, 5.3) | 3.8 (2.5,7.1) | 3.4 (2.1, 5.2) | 0.309 |
| Monocyte count, *10^9/L* | 0.75 (0.5, 1.2) | 0.73 (0.46, 1.25) | 0.75 (0.49, 1.2) | 0.922 |
| **Hemoglobin, g/dl** | 10.5 (8.1,11.6) | 8.6 (6.5, 10) | 10.7 (8.4, 11.9) | < **0.001** |
| WBC, *10^9/L* | 8.5 (5.2, 12.3) | 11.1 (5.9, 14.2) | 9.6 (6.4, 14.1) | 0.455 |

*Common comorbidities included Cerebral palsy, sickle cell disease, malaria, oral candidiasis; WBC, white blood cell count

comorbidities (51.5% vs. 22.4%). A higher proportion of patients who had TB had lower hemoglobin levels as compared to those without TB (8.6 vs. 10.7 g/dl).

## Regression analysis of factors associated with TB

**Table 2** shows results of univariable analysis and multivariable analysis of factors associated with TB. At univariable analysis, patients from rural areas were 1.78 times more likely to have TB compared to those living in urban areas. People living with HIV were 12.8 times more likely to have TB than those who were HIV negative. Patients who had malnutrition had 15.76 times more likely of having TB compared to those who did not have malnutrition. Patients who had comorbidities were 3.68 times more likely to develop TB as opposed to those without comorbidities. Weight and height were negatively associated with TB.

At multivariable analysis, People living with HIV and those who had malnutrition were 6.30 times and 10.38 more likely to have TB compared to those who were HIV negative and without malnutrition, respectively.

## Discussion

The objective of this study was twofold: firstly, to determine the prevalence of active tuberculosis disease among hospitalized children under the age of 15 years old, and secondly, to identify

**Table 2. Univariable and multivariable logistic regression of factors associated with tuberculosis infection.**

| | OR (95%) | P-value | AOR (95%, Cl) | P-Value |
|---|---|---|---|---|
| **Residence** | | | | |
| Urban | 1 | | 1 | |
| Rural | 1.78(1.07,2.96) | **0.026** | 1.91(0.90, 4.03) | 0.090 |
| **HIV Status** | | | | |
| HIV negative | 1 | | 1 | |
| Living with HIV | 12.8 (7.03, 23.37) | <**0.001** | 6.30 (2.85,13.89) | < **0.001** |
| **Malnutrition** | | | | |
| No | 1 | | 1 | |
| Yes | 15.76 (8.83, 28.13) | <**0.001** | 10.38(4.78,22.55) | < **0.001** |
| **Comorbidities** | | | | |
| No | 1 | | | |
| Yes | 3.68 (2.19,6.17) | <**0.001** | | |
| **Haemoglobin** | 0.84(0.76, 0.92) | <**0.001** | 0.89 (0.77, 1.01) | 0.067 |
| **Anthropometric measurement** | | | | |
| Weight, *kgs* | 0.95 (0.91, 0.99) | < **0.011** | | |
| Height, cm | 0.99(0.98, 0.99) | <**0.009** | | |

sociodemographic and clinical factors associated with active tuberculosis disease at LUTH pediatric department.

Our study found a 9.4% prevalence of active TB among hospitalized children under the age of 15 years at LUTH Pediatric department. This aligns with findings from a 2016 study in Zambia, which showed a TB prevalence of 8% among children under 15 years old who died in inpatient wards, and a national prevalence of 9% among children [9, 12]. The true burden of childhood TB at our facility is likely much higher. Limited availability of advanced diagnostics, like Xpert MTB/RIF, hinders accurate diagnosis, leading to low notification rates [13, 14]. Other contributing factors include non-specific symptoms mimicking other diseases in children, limited healthcare provider expertise in diagnosing these cases, and lack of public awareness in the community [13, 14].

Our study found that among hospitalized children, those living with HIV were more likely to have TB compared to those without HIV. This finding is consistent with other research which have demonstrated a synergistic relationship between HIV and respiratory infection like TB, where HIV infection increases the risk of TB acquisition and accelerates the progression of TB disease because it leads to immune deficiency [15, 16].

Malnutrition also emerged as a significant associated factor for TB among hospitalized pediatric patients in this study. Malnutrition weakens the immune system, making individuals more susceptible to TB and increasing the risk of tuberculosis disease progression [17]. Addressing child malnutrition which is high in our setting through nutritional interventions and supplementation may have a dual benefit in reducing the burden of TB among pediatric populations [18, 19].

While our study provides valuable insights into the burden of TB among hospitalized pediatric populations, it is important to acknowledge its limitations. Being a retrospective cross-sectional study, it may not capture all associations, and poor record-keeping hindered extraction of additional data or factors known to be associated with TB such as housing condition, information on TB contact by the child. Second, this study only included hospitalized patients, excluding children treated in the outpatient department. This limitation might overestimate the prevalence of TB disease among children seen at the LUTH Pediatrics Department. Despite

these constraints, our findings offer a basis for further research in this area, emphasizing the need for longitudinal studies to better understand TB dynamics among children.

## Conclusion

Our study findings reveal a high burden of tuberculosis among children hospitalized under 15 years old at our facility. Notably, HIV status, and malnutrition significantly correlate with tuberculosis infection. To mitigate this burden effectively, pediatric tuberculosis control strategies must prioritize addressing HIV and malnutrition.

## Supporting information

**S1 Checklist. STROBE statement—checklist of items that should be included in reports of *cross-sectional studies.***
(DOCX)

**S1 Dataset.**
(XLSX)

## Acknowledgments

The authors would like to thank Livingstone University Teaching hospital management for having granted permission to conduct the study at the pediatric department.

## Author Contributions

**Conceptualization:** Lukundo Siame, Gift C. Chama, Sepiso K. Masenga.

**Data curation:** Lukundo Siame, Gift C. Chama, Sepiso K. Masenga.

**Formal analysis:** Lukundo Siame, Sepiso K. Masenga.

**Investigation:** Lukundo Siame, Sepiso K. Masenga.

**Methodology:** Lukundo Siame, Sepiso K. Masenga.

**Project administration:** Sepiso K. Masenga.

**Supervision:** Sepiso K. Masenga.

**Validation:** Sepiso K. Masenga.

**Visualization:** Lukundo Siame, Sepiso K. Masenga.

**Writing – original draft:** Lukundo Siame, Gift C. Chama, Sepiso K. Masenga.

**Writing – review & editing:** Lukundo Siame, Gift C. Chama, Sepiso K. Masenga.

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
