## [Decision Letter · Decision Letter 0]

10 Jun 2024

PONE-D-24-10233Correlates of Tuberculosis in children under the age of 15 yearsPLOS ONE

Dear Dr. Masenga,

Thank you for submitting your manuscript to PLOS ONE. After careful consideration, we feel that it has merit but does not fully meet PLOS ONE’s publication criteria as it currently stands. Therefore, we invite you to submit a revised version of the manuscript that addresses the points raised during the review process. To aid the review process, kindly use line numbering in the revised manuscript in your subsequent submission.Please submit your revised manuscript by Jul 25 2024 11:59PM. If you will need more time than this to complete your revisions, please reply to this message or contact the journal office at plosone@plos.org. Please include the following items when submitting your revised manuscript:A rebuttal letter that responds to each point raised by the academic editor and reviewer(s). You should upload this letter as a separate file labeled 'Response to Reviewers'.A marked-up copy of your manuscript that highlights changes made to the original version. You should upload this as a separate file labeled 'Revised Manuscript with Track Changes'.An unmarked version of your revised paper without tracked changes. You should upload this as a separate file labeled 'Manuscript'.

We look forward to receiving your revised manuscript.

Kind regards,

Mobolanle Balogun

Academic Editor

PLOS ONE

Journal Requirements:

3. We note you have included a table to which you do not refer in the text of your manuscript. Please ensure that you refer to Table 2 in your text; if accepted, production will need this reference to link the reader to the Table.

Reviewers' comments:

Reviewer's Responses to Questions

**Comments to the Author**

1. Is the manuscript technically sound, and do the data support the conclusions?

Reviewer #1: Yes

Reviewer #2: Yes

2. Has the statistical analysis been performed appropriately and rigorously? 

Reviewer #1: No

Reviewer #2: Yes

3. Have the authors made all data underlying the findings in their manuscript fully available?

Reviewer #1: Yes

Reviewer #2: Yes

4. Is the manuscript presented in an intelligible fashion and written in standard English?

Reviewer #1: Yes

Reviewer #2: Yes

5. Review Comments to the Author

Reviewer #1: It would be helpful to have line numbers for ease of reference when reviewing the manuscript.

1. This sentence in the introduction is not accurate and must be revised “According to the World Health Organization (WHO), an estimated 10 million people worldwide were infected with tuberculosis in 2020” , It developed and not infected. Almost 1/3 of the global population are infected with TB.

2. Please check reference 5, I don't think it is the correct reference. Going through the artical, there is no indication or discussion about prevalence of TB. Also, prevalence of TB of 12% and 15% in Malawi and South Africa seems very high. Check that you have the correct interpretation of this. Many high burden countries have a prevalence of around 1% of TB, it is lower in children. Perhaps the authors wanted to say that “out of all TB notification, children accounted for 12 and 15% etc”

3. The reference for the first sentence in paragraph 3 of the introduction session is not the best reference for this and it is old (2016). There are newer references for this. For example WHO list of high burden countries.....

4. The first sentence of paragraph 3 needs to be revised for clarity based on the list of high burden countries

5. Consider revising this sentence “ A recent national survey conducted by kapata et al (2016) estimated the prevalence of 455/ 100000 persons for a population aged 15 years and older and the burden was high among people living with HIV (PLWH) (10)”

This survey was conducted in 2013/ 2014, publication was 2016, therefore it is not recent, perhaps, the last survey conducted in xx years found .....

6. What guided or what assumptions were used to set 50% prevalence of TB? That is half the population with TB. Did the authors get help from an epidemiologist or a biostastician to do the sample size calculations

7. The authors tend to mix tuberculosis infection with active TB disease, this needs to be corrected through the manuscript. Treatment for tuberculosis infection is not the same as treatment for active TB disease.

8. Clarify if “Data were collected by three (3) trained research assistants who abstracted data from patient files between August 1, 2023 and January 8th, 2024.” this means only for patients treated during this time period were sampled. And what guided the selection of these dates.

9. Data analysis- more detail is needed on how regression analysis was done, specially how the model was built to decide on the final variables for inclusion in the model.

Results section

10. Why use the word about, when you have an actual measure “ about 9.2 people were living with HIV.....”

11. Table 1 is a bot confusing. Suggest to present column % in columns 3 and 4 and not row %s

Discussion

12. First line in the discussion is again mixing up TB infection and TB disease

13. Second paragraph 1st sentence needs to be specific to indicate that the finding is among hospitalised children. As it is written, it give the impression that this finding is in the general population.

14. For this sentence “This prevalence rate is slightly different with previous studies conducted in Sudan which was 15.3% and other studies have estimated it to be approximately 11% (12,13).” Check and be sure you are comparing similar populations. i.e hospitalised children

15. In the discussion, you might want to also reference the Zambian postmortem studies by Chintu et al, Mudenda et al

Reviewer #2: please state How many years record was reviewed

Was the data a clinic data , admissions or total number of children seen in the hospital and over what specified period ?

The mauscript is relevant , but they need to state how many years data was the review ie period

Percentage of HIV 9.1 % or 9 %

LIMITATION. One of the limitation the number of children of 66 out of 700 with tuberculosis was small and diificult to interprete , however this is possilby how many were avilaible to them , they need to state where the patient were from ,refered or patient within the community ie scope of the centre where the study was carried out

6. PLOS authors have the option to publish the peer review history of their article (what does this mean?). If published, this will include your full peer review and any attached files.

Reviewer #1: No

Reviewer #2: **Yes: **Patrica Eyanya Akintan

---

## [Author Response · Author response to Decision Letter 0]

20 Jun 2024

RESPONSES TO EDITORIAL AND REVIEWER COMMENTS

Response: done

Response: done

3. We note you have included a table to which you do not refer in the text of your manuscript. Please ensure that you refer to Table 2 in your text; if accepted, production will need this reference to link the reader to the Table.

Response: done

Response: done

Reviewer #1: It would be helpful to have line numbers for ease of reference when reviewing the manuscript.

Response: thank very for the suggestion. We have added line numbers to the manuscript. 

1. This sentence in the introduction is not accurate and must be revised “According to the World Health Organization (WHO), an estimated 10 million people worldwide were infected with tuberculosis in 2020”, It developed and not infected. Almost 1/3 of the global population are infected with TB.

Response: thank you very for the correction. The sentence has been revised to imply development of cases instead of infections. 

2. Please check reference 5, I don't think it is the correct reference. Going through the article, there is no indication or discussion about prevalence of TB. Also, prevalence of TB of 12% and 15% in Malawi and South Africa seems very high. Check that you have the correct interpretation of this. Many high burden countries have a prevalence of around 1% of TB, it is lower in children. Perhaps the authors wanted to say that “out of all TB notification, children accounted for 12 and 15% etc”

Response: thank very for the correction. The sentence has been revised with new accurate information. 

3. The reference for the first sentence in paragraph 3 of the introduction session is not the best reference for this and it is old (2016). There are newer references for this. For example WHO list of high burden countries?

Response: thank very for the correction. The sentence has been revised with latest data from WHO report of 2023 Global tuberculosis report 2023.

4. The first sentence of paragraph 3 needs to be revised for clarity based on the list of high burden countries

Response: thank very, the sentence has been revised and clarified 

5. Consider revising this sentence “A recent national survey conducted by kapata et al (2016) estimated the prevalence of 455/ 100000 persons for a population aged 15 years and older and the burden was high among people living with HIV (PLWH) (10)”

This survey was conducted in 2013/ 2014, publication was 2016, therefore it is not recent, perhaps, the last survey conducted in xx years found.

Response: thank very for the suggestion. The sentence has been updated with current incidence as of 2022 published by Health press bullet in 2022 and WHO global tuberculosis report 2023. 

6. What guided or what assumptions were used to set 50% prevalence of TB? That is half the population with TB. Did the authors get help from an epidemiologist or a biostastician to do the sample size calculations

Response: thank you very much. We originally used 50% in order to generate the highest maximal sample size so that our study is highly powered. However, we have now recast and recalculated the sample size with help from a biostatistician. We have used a prevalence of 9 % of active TB case recorded in Zambia. Our study is still highly powered with a current sample size above the minimal required. We thank the reviewer for highlighting this.

7. The authors tend to mix tuberculosis infection with active TB disease, this needs to be corrected through the manuscript. Treatment for tuberculosis infection is not the same as treatment for active TB disease.

Response: thank very. We have modified this throughput the manuscript.

8. Clarify if “Data were collected by three (3) trained research assistants who abstracted data from patient files between August 1, 2023 and January 8th, 2024.” this means only for patients treated during this time period were sampled. And what guided the selection of these dates.

Response: thank very. “Data were collected by three (3) trained research assistants who abstracted data from patient files between August 1, 2023 and January 8th, 2024.” Was meant to show the period when data abstraction was done. We abstracted children’s files who were treated between 1st January 2020 to 31 December 2022. We have now clarified this in the manuscript’ methods section. Thank you for bringing up this.

9. Data analysis- more detail is needed on how regression analysis was done, especially how the model was built to decide on the final variables for inclusion in the model.

Response: thank very. We have clarified how the model was built. The regression was built based on the variables which had a p value less than 0.05. 

Results section

10. Why use the word about, when you have an actual measure “about 9.2 people were living with HIV.....”

Response: thank very. The sentence has been corrected.

11. Table 1 is a bit confusing. Suggest to present column % in columns 3 and 4 and not row %s

Response: thank very for the suggestion. Column % have now been used in table one. 

Discussion

12. First line in the discussion is again mixing up TB infection and TB disease

Response: thank very much for the correction. We have now corrected this and all other instances where we mixed up TB infection and TB disease

13. Second paragraph 1st sentence needs to be specific to indicate that the finding is among hospitalised children. As it is written, it gives the impression that this finding is in the general population.

Response: thank very for the suggestion. We have corrected

14. For this sentence “This prevalence rate is slightly different with previous studies conducted in Sudan which was 15.3% and other studies have estimated it to be approximately 11% (12,13).” Check and be sure you are comparing similar populations. i.e hospitalized children

Response: thank you very much for the correction. The studies have been changed to reflect the populations being compared 

15. In the discussion, you might want to also reference the Zambian postmortem studies by Chintu et al, Mudenda et al

Response: thank very for the suggestion. The study of mudenda et al has been cited as it was conducted in-patients 

REVIEWER #2: 

please state How many years record was reviewed

Response: thank very. The data reviewed was a period of 3 years (between 1st January 2020 to 31 December 2022) we have now included this in the design and setting of the methods section. 

Was the data a clinic data, admissions or total number of children seen in the hospital and over what specified period?

Response: thank very. The data was admission of children over a period of 3 years (between 1st January 2020 to 31 December 2022) we have included in the design and setting in the methods section. 

The manuscript is relevant, but they need to state how many years data was the review ie period

Percentage of HIV 9.1 % or 9 %

Response: thank very. The number of years data was reviewed has been stated now in the methods. The prevalence of HIV among the children was 9.2% (62/675). 

LIMITATION. One of the limitation the number of children of 66 out of 700 with tuberculosis was small and diificult to interprete , however this is possibly how many were available to them , they need to state where the patient were from ,refered or patient within the community ie scope of the center where the study was carried out

Response: thank very. We have included your suggestion as one of the limitations and also included details of the study population

---

## [Decision Letter · Decision Letter 1]

15 Aug 2024

PONE-D-24-10233R1Correlates of Tuberculosis in children under the age of 15 yearsPLOS ONE

Dear Dr. Masenga,

Thank you for submitting your manuscript to PLOS ONE. After careful consideration, we feel that it has merit but does not fully meet PLOS ONE’s publication criteria as it currently stands. Therefore, we invite you to submit a revised version of the manuscript that addresses the points raised during the review process.

We look forward to receiving your revised manuscript.

Kind regards,

Mobolanle Balogun

Academic Editor

PLOS ONE

Journal Requirements:

**Additional Editor Comments:**

Thank you for attending to most of the reviewers' comments. Please attend to the remaining concerns particularly about sample size estimation and interpretation of your results. Please provide a clear and detailed rebuttal where necessary.

Reviewers' comments:

Reviewer's Responses to Questions

**Comments to the Author**

1. If the authors have adequately addressed your comments raised in a previous round of review and you feel that this manuscript is now acceptable for publication, you may indicate that here to bypass the “Comments to the Author” section, enter your conflict of interest statement in the “Confidential to Editor” section, and submit your "Accept" recommendation.

Reviewer #1: (No Response)

2. Is the manuscript technically sound, and do the data support the conclusions?

Reviewer #1: Partly

3. Has the statistical analysis been performed appropriately and rigorously? 

Reviewer #1: No

4. Have the authors made all data underlying the findings in their manuscript fully available?

Reviewer #1: Yes

5. Is the manuscript presented in an intelligible fashion and written in standard English?

Reviewer #1: Yes

6. Review Comments to the Author

Reviewer #1: Line 1:

Use updated TB report. The latest is 2022

Line 94:

Sample size calculation is still an issue. I think the authors are using the wrong reference for this. In what population is this estimate from? It is incorrect to use the proportion of children notified as the prevalence. I think this is what the authors did but I may be wrong. The authors needed to have used any data available on prevalence of TB among hospitalised children. What about postmortem studies from Zambia, what did that show.

Line 142 Regression analysis, further description is needed here. How was the final model arrived at? Was it backward or forward addition?

Why are the authors reporting unadjusted ORs. Unj OR can be due to chance occurrence? Adjusted ORs is what is reported. So only HIV and Malnutrition were associated with TB. But the authors need to advise how the regression analysis was done.

Line 232: The authors need to be clear that they are referring to hospitalised children and should not generalised this finding. This is a highly selected group so this sentence needs to mention this.

Line 238 perhaps overestimate?

Conclusion: Is generalised. Needs to be specific that this is in hospitalised children.

7. PLOS authors have the option to publish the peer review history of their article (what does this mean?). If published, this will include your full peer review and any attached files.

Reviewer #1: No

---

## [Author Response · Author response to Decision Letter 1]

15 Sep 2024

15th September 2024

To the Editor,

Dear Editor,

Ref: Reviewer responses 

The above matter refers. 

We would like to thank the reviewer for their invaluable suggestions to further improve our manuscript. We have revised the manuscript accordingly. We hope it is now satisfactory. Below are the specific changes and responses.

Reviewer #1: Line 1:

Use updated TB report. The latest is 2022

Response: thank you for the correction, the statistics have been updated as suggested

Line 94:

Sample size calculation is still an issue. I think the authors are using the wrong reference for this. In what population is this estimate from? It is incorrect to use the proportion of children notified as the prevalence. I think this is what the authors did but I may be wrong. The authors needed to have used any data available on prevalence of TB among hospitalised children. What about postmortem studies from Zambia, what did that show.

Response: thank you very much the suggestion. We have recalculated the sample size using the 2016 bates et al postmortem studies among hospitalized pediatric patients who died. Which revised the prevalence to 8 % active TB. 

Line 142 Regression analysis, further description is needed here. How was the final model arrived at? Was it backward or forward addition? Why are the authors reporting unadjusted ORs. Unj OR can be due to chance occurrence? Adjusted ORs is what is reported. So only HIV and Malnutrition were associated with TB. But the authors need to advise how the regression analysis was done.

Response: Thank you very much. We have reanalyzed the data using the forward stepwise Generalized Linear Models equations (GLM) via the logit link function to account for correlation structure within observations and to select the most significant predictors that were included in the final model. We first reported the unadjusted ORs just to show that it was the first step we did to build the model; however, the unadjusted ORs were not discussed in the discussion due to potential confounders. 

Line 232: The authors need to be clear that they are referring to hospitalised children and should not generalised this finding. This is a highly selected group so this sentence needs to mention this.

Response: we have corrected as suggested. Thank you

Conclusion: Is generalized. Needs to be specific that this is in hospitalized children.

Response: thank you for the correction. The sentence has been corrected 

Line 238 perhaps overestimate? 

Response: thank you. We have substituted underestimate with overestimate as suggested.

Conclusion: Is generalized. Needs to be specific that this is in hospitalized children.

Response: thank you for the concern. We have now specified the population is among hospitalized children at our facility 

Once again, we thank you for your support in helping to improve our manuscript 

We look forward to hearing from you at your earliest convenience. 

Yours sincerely,

Prof. Sepiso K. Masenga

---

## [Editor Report · Decision Letter 2]

10 Oct 2024

Correlates of Tuberculosis in children under the age of 15 years

PONE-D-24-10233R2

Dear Dr. Masenga,

We’re pleased to inform you that your manuscript has been judged scientifically suitable for publication and will be formally accepted for publication once it meets all outstanding technical requirements.

Kind regards,

Mobolanle Balogun

Academic Editor

PLOS ONE
---

## [Editor Report · Acceptance letter]

18 Nov 2024

PONE-D-24-10233R2 

PLOS ONE

Dear Dr. Masenga, 

I'm pleased to inform you that your manuscript has been deemed suitable for publication in PLOS ONE. Congratulations! Your manuscript is now being handed over to our production team.

Kind regards, 

on behalf of

Dr. Mobolanle Balogun 

Academic Editor

PLOS ONE